# Blockwise Parallel Decoding for Deep Autoregressive Models

**Mitchell Stern**[*]
University of California, Berkeley
mitchell@berkeley.edu

**Noam Shazeer**
Google Brain
noam@google.com

**Jakob Uszkoreit**
Google Brain
usz@google.com

## Abstract

Deep autoregressive sequence-to-sequence models have demonstrated impressive performance across a wide variety of tasks in recent years. While common architecture classes such as recurrent, convolutional, and self-attention networks make different trade-offs between the amount of computation needed per layer and the length of the critical path at training time, generation still remains an inherently sequential process. To overcome this limitation, we propose a novel blockwise parallel decoding scheme in which we make predictions for multiple time steps in parallel then back off to the longest prefix validated by a scoring model. This allows for substantial theoretical improvements in generation speed when applied to architectures that can process output sequences in parallel. We verify our approach empirically through a series of experiments using state-of-the-art self-attention models for machine translation and image super-resolution, achieving iteration reductions of up to 2x over a baseline greedy decoder with no loss in quality, or up to 7x in exchange for a slight decrease in performance. In terms of wall-clock time, our fastest models exhibit real-time speedups of up to 4x over standard greedy decoding.

## 1 Introduction

Neural autoregressive sequence-to-sequence models have become the de facto standard for a wide variety of tasks, including machine translation, summarization, and speech synthesis (Vaswani et al., 2017; Rush et al., 2015; van den Oord et al., 2016). One common feature among recent architectures such as the Transformer and convolutional sequence-to-sequence models is an increased capacity for parallel computation, making them a better fit for today's massively parallel hardware accelerators (Vaswani et al., 2017; Gehring et al., 2017). While advances in this direction have allowed for significantly faster training, outputs are still generated one token at a time during inference, posing a substantial challenge for many practical applications (Oord et al., 2017).

In light of this limitation, a growing body of work is concerned with different approaches to accelerating generation for autoregressive models. Some general-purpose methods include probability density distillation (Oord et al., 2017), subscaling (Kalchbrenner et al., 2018), and decomposing the problem into the autoregressive generation of a short sequence of discrete latent variables followed by a parallel generation step conditioned on the discrete latents (Kaiser et al., 2018). Other techniques are more application-specific, such as the non-autoregressive Transformer for machine translation (Gu et al., 2018). While speedups of multiple orders of magnitude have been achieved on tasks with high output locality like speech synthesis, to the best of our knowledge, published improvements in machine translation either show much more modest speedups or come at a significant cost in quality.

---

[*]Work performed while the author was an intern at Google Brain.

In this work, we propose a simple algorithmic technique that exploits the ability of some architectures, such as the Transformer (Vaswani et al., 2017), to score all output positions in parallel. We train variants of the autoregressive model to make predictions for multiple future positions beyond the next position modeled by the base model. At test time, we employ these proposal models to independently and in parallel make predictions for the next several positions. We then determine the longest prefix of these predictions that would have generated under greedy decoding by scoring each position in parallel using the base model. If the length of this prefix is greater than one, we are able to skip one or more iterations of the greedy decoding loop.

In our experiments, our technique approximately doubles generation speed at no loss in quality relative to greedy decoding from an autoregressive model. Together with knowledge distillation and approximate decoding strategies, we can increase the speedup in terms of decoding iterations to up to five-fold at a modest sacrifice in quality for machine translation and seven-fold for image super-resolution. These correspond to wall-clock speedups of three-fold and four-fold, respectively.

In contrast to the other previously mentioned techniques for improving generation speed, our approach can furthermore be implemented on top of existing models with minimal modifications. Our code is publicly available in the open-source Tensor2Tensor library (Vaswani et al., 2018).

## 2   Greedy Decoding

In a sequence-to-sequence problem, we are given an input sequence $x = (x_1, \ldots, x_n)$, and we would like to predict the corresponding output sequence $y = (y_1, \ldots, y_m)$. These sequences might be source and target sentences in the case of machine translation, or low-resolution and high-resolution images in the case of image super-resolution. One common approach to this problem is to learn an autoregressive scoring model $p(y \mid x)$ that decomposes according to the left-to-right factorization

$$\log p(y \mid x) = \sum_{j=0}^{m-1} \log p(y_{j+1} \mid y_{\leq j}, x).$$

The inference problem is then to find $y^* = \operatorname{argmax}_y p(y \mid x)$.

Since the output space is exponentially large, exact search is intractable. As an approximation, we can perform greedy decoding to obtain a prediction $\hat{y}$ as follows. Starting with an empty sequence $\hat{y}$ and $j = 0$, we repeatedly extend our prediction with the highest-scoring token $\hat{y}_{j+1} = \operatorname{argmax}_{y_{j+1}} p(y_{j+1} \mid \hat{y}_{\leq j}, x)$ and set $j \leftarrow j + 1$ until a termination condition is met. For language generation problems, we typically stop once a special end-of-sequence token has been generated. For image generation problems, we simply decode for a fixed number of steps.

## 3   Blockwise Parallel Decoding

Standard greedy decoding takes $m$ steps to produce an output of length $m$, even for models that can efficiently score sequences using a constant number of sequential operations. While brute-force enumeration of output extensions longer than one token is intractable when the size of the vocabulary is large, we can still attempt to exploit parallelism within the model by training a set of auxiliary models to propose candidate extensions.

Let the original model be $p_1 = p$, and suppose that we have also learned a collection of auxiliary models $p_2, \ldots, p_k$ for which $p_i(y_{j+i} \mid y_{\leq j}, x)$ is the probability of the $(j + i)$th token being $y_{j+i}$ given the first $j$ tokens. We propose the following blockwise parallel decoding algorithm (illustrated in Figure 1), which is guaranteed to produce the same prediction $\hat{y}$ that would be found under greedy decoding but uses as few as $m/k$ steps. As before, we start with an empty prediction $\hat{y}$ and set $j = 0$. Then we repeat the following three substeps until the termination condition is met:

- **Predict:** Get the block predictions $\hat{y}_{j+i} = \operatorname{argmax}_{y_{j+i}} p_i(y_{j+i} \mid \hat{y}_{\leq j}, x)$ for $i = 1, \ldots, k$.

- **Verify:** Find the largest $\hat{k}$ such that $\hat{y}_{j+i} = \operatorname{argmax}_{y_{j+i}} p_1(y_{j+i} \mid \hat{y}_{\leq j+i-1}, x)$ for all $1 \leq i \leq \hat{k}$. Note that $\hat{k} \geq 1$ by the definition of $\hat{y}_{j+1}$.

- **Accept:** Extend $\hat{y}$ with $\hat{y}_{j+1}, \ldots, \hat{y}_{j+\hat{k}}$ and set $j \leftarrow j + \hat{k}$.

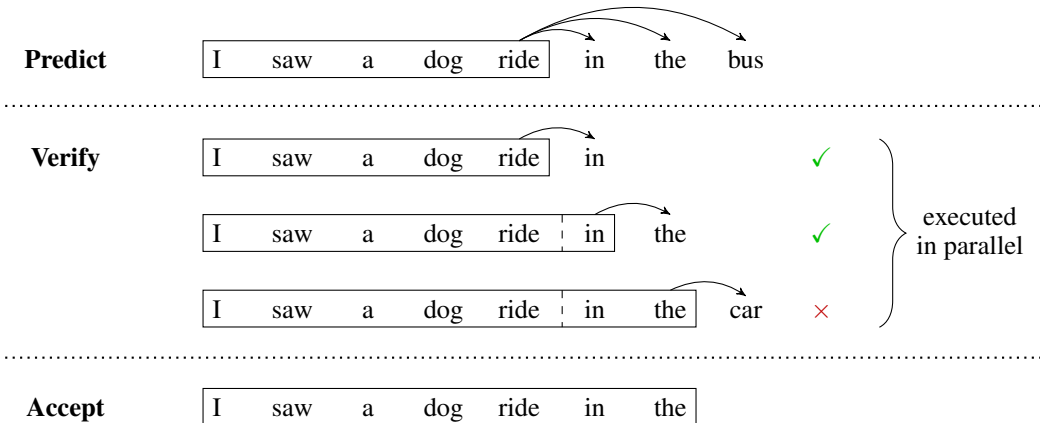

Figure 1: The three substeps of blockwise parallel decoding. In the **predict** substep, the greedy model and two proposal models independently and in parallel predict "in", "the", and "bus". In the **verify** substep, the greedy model scores each of the three independent predictions, conditioning on the previous independent predictions where applicable. When using a Transformer or convolutional sequence-to-sequence model, these three computations can be done in parallel. The highest-probability prediction for the third position is "car", which differs from the independently predicted "bus". In the **accept** substep, $\hat{y}$ is hence extended to include only "in" and "the" before making the next $k$ independent predictions.

In the **predict** substep, we find the local greedy predictions of our base scoring model $p_1$ and the auxiliary proposal models $p_2, \ldots, p_k$. Since these are disjoint models, each prediction can be computed in parallel, so there should be little time lost compared to a single greedy prediction.

Next, in the **verify** substep, we find the longest prefix of the proposed length-$k$ extension that would have otherwise been produced by $p_1$. If the scoring model can process this sequence of $k$ tokens in fewer than $k$ steps, this substep will help save time overall provided more than one token is correct.

Lastly, in the **accept** substep, we extend our hypothesis with the verified prefix. By stopping early if the base model and the proposal models start to diverge in their predictions, we ensure that we will recover the same output that would have been produced by running greedy decoding with $p_1$.

The potential of this scheme to improve decoding performance hinges crucially on the ability of the base model $p_1$ to execute all predictions made in the **verify** substep in parallel. In our experiments we use the Transformer model (Vaswani et al., 2017). While the total number of operations performed during decoding is quadratic in the number of predictions, the number of necessarily sequential operations is constant regardless of output length. This allows us to execute the **verify** substep for a number of positions in parallel without spending additional wall-clock time.

## 4    Combined Scoring and Proposal Model

When using a Transformer for scoring, the version of our algorithm presented in Section 3 requires two model invocations per step: one parallel invocation of $p_1, \ldots, p_k$ in the prediction substep, and an invocation of $p_1$ in the verification substep. This means that even with perfect auxiliary models, we will only reduce the number of model invocations from $m$ to $2m/k$ instead of the desired $m/k$.

As it turns out, we can further reduce the number of model invocations from $2m/k$ to $m/k + 1$ if we assume a combined scoring and proposal model, in which case the $n$th verification substep can be merged with the $(n + 1)$st prediction substep.

More specifically, suppose we have a single Transformer model which during the verification substep computes $p_i(y_{j+i'+i} \mid \hat{y}_{\leq j+i'}, x)$ for all $i = 1, \ldots, k$ and $i' = 1, \ldots, k$ in a constant number of operations. This can be implemented for instance by increasing the dimensionality of the final projection layer by a factor of $k$ and computing $k$ separate softmaxes per position. Invoking the model after plugging in the $k$ future predictions from the prediction substep yields the desired outputs.

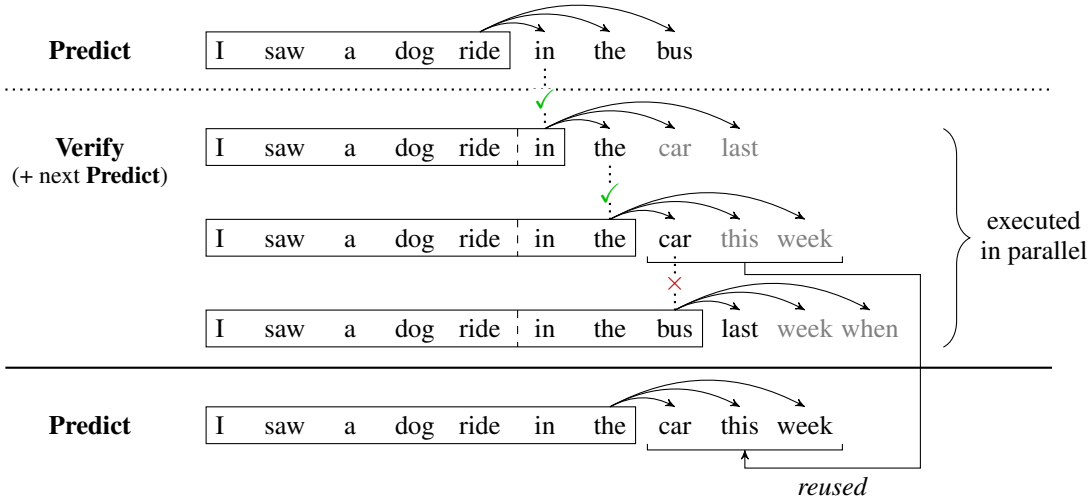

Figure 2: Combining the scoring and proposal models allows us to merge the previous verification substep with the next prediction substep. This makes it feasible to call the model just once per iteration rather than twice, halving the number of model invocations required for decoding.

Under this setup, after $\hat{k}$ has been computed during verification, we will have already computed $p_i(y_{j+\hat{k}+i} \mid y_{\leq j+\hat{k}}, x)$ for all $i = 1, \ldots, k$, which is exactly what is required for the prediction substep in the next iteration of decoding. Hence these substeps can be merged together, reducing the number of model invocations by a factor of two for all but the very first iteration.

Figure 2 illustrates the process. Note that while proposals have to be computed for every position during the verification substep, all predictions can still be made in parallel.

## 5   Approximate Inference

The approach to block parallel decoding we have described so far produces the same output as a standard greedy decode. By relaxing the criterion used during verification, we can allow for additional speedups at the cost of potentially deviating from the greedy output.

### 5.1   Top-$k$ Selection

Rather than requiring that a prediction exactly matches the scoring model's prediction, we can instead ask that it lie within the top $k$ items. To accomplish this, we replace the verification criterion with

$$\hat{y}_{j+i} \in \text{top-}k_{y_{j+i}} p_1(y_{j+i} \mid \hat{y}_{\leq j+i-1}, x).$$

### 5.2   Distance-Based Selection

In problems where the output space admits a natural distance metric $d$, we can replace the exact match against the highest-scoring element with an approximate match:

$$d\left(\hat{y}_{j+i}, \operatorname*{argmax}_{y_{j+i}} p_1(y_{j+i} \mid \hat{y}_{\leq j+i-1}, x)\right) \leq \epsilon.$$

In the case of image generation, we let $d(u, v) = |u - v|$ be the absolute difference between intensities $u$ and $v$ within a given color channel.

### 5.3   Minimum Block Size

It is possible that the first non-greedy prediction within a given step is incorrect, in which case only a single token would be added to the hypothesis. To ensure a minimum speedup, we could require that

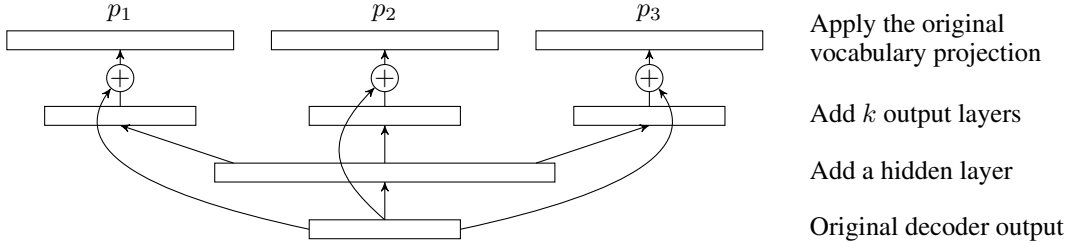

| | |
|---|---|
| $p_1$    $p_2$    $p_3$ | Apply the original vocabulary projection |
| | Add $k$ output layers |
| | Add a hidden layer |
| | Original decoder output |

Figure 3: The modification we make to a Transformer to obtain a combined scoring and prediction model. To make predictions for the next $k$ positions instead of one position, we insert a multi-output feedforward layer with residual connections after the original decoder output layer, then apply the original vocabulary projection to all outputs.

at least $1 < \ell \leq k$ tokens be added during each decoding step. Setting $\ell = k$ would correspond to parallel decoding with blocks of fixed size $k$.

# 6 Implementation and Training

We implement the combined scoring and proposal model described in Section 4 for our experiments. Given a baseline Transformer model pre-trained for a given task, we insert a single feedforward layer with hidden size $k \times d_{\text{hidden}}$ and output size $k \times d_{\text{model}}$ between the decoder output and the final projection layer, where $d_{\text{hidden}}$ and $d_{\text{model}}$ are the same layer dimensions used in the rest of the network. A residual connection between the input and each of the $k$ outputs is included. The original projection layer is identically applied to each of the $k$ outputs to obtain the logits for $p_1, \ldots, p_k$. See Figure 3 for an illustration.

Due to memory constraints at training time, we are unable to use the mean of the $k$ cross entropy losses corresponding to $p_1, \ldots, p_k$ as the overall loss. Instead, we select one of these sub-losses uniformly at random for each minibatch to obtain an unbiased estimate of the full loss. At inference time, all logits can be computed in parallel with marginal cost relative to the base model.

## 6.1 Fine Tuning

An important question is whether or not the original parameters of the pre-trained model should be fine tuned for the modified joint prediction task. If they are kept frozen, we ensure that the quality of the original model is retained, perhaps at the cost of less accurate future prediction. If they are fine tuned, we might improve the model's internal consistency but suffer a loss in terms of final performance. We investigate both options in our experiments.

## 6.2 Knowledge Distillation

The practice of knowledge distillation (Hinton et al., 2015; Kim and Rush, 2016), in which one trains a model on the outputs of another model, has been shown to improve performance on a variety of tasks, potentially even when teacher and student models have the same architecture and model size (Furlanello et al., 2017). We posit that sequence-level distillation could be especially useful for blockwise parallel decoding, as it tends to result in a training set with greater predictability due to consistent mode breaking from the teacher model. For our language task, we perform experiments using both the original training data and distilled training data to determine the extent of the effect. The distilled data is produced via beam decoding using a pre-trained model with the same hyperparameters as the baseline but a different random seed. The beam search hyperparameters are those from Vaswani et al. (2017).

# 7 Experiments

We implement all our experiments using the open-source Tensor2Tensor framework (Vaswani et al., 2018). Our code is publicly available within this same library.

### 7.1 Machine Translation

For our machine translation experiments, we use the WMT 2014 English-German translation dataset. Our baseline model is a Transformer trained for 1,000,000 steps on 8 P100 GPUs using the `transformer_base` hyperparameter set in Tensor2Tensor. Using greedy decoding, it attains a BLEU score of 25.56 on the newstest2013 development set.

On top of this, we train a collection of combined scoring and proposal Transformer models for various block sizes $k$; see Section 6 for implementation details. Each model is trained for an additional 1,000,000 steps on the same hardware, either on the original training data or on distilled data obtained from beam search predictions from a separate baseline run. Optimizer accumulators for running averages of first and second moments of the gradient are reset for the new training runs, as is the learning rate schedule.

We measure the BLEU score and the mean accepted block size $\hat{k}$ on the development set under a variety of settings. Results are reported in Table 1.[1]

| $k$ | Regular ● | Distillation ■ | Fine Tuning ▲ | Both ◆ | |
|-----|-----------|----------------|---------------|--------|---|
| 1 | 26.00 / 1.00 | 26.41 / 1.00 | | | |
| 2 | 25.81 / 1.51 | 26.52 / 1.55 | 25.74 / 1.78 | 26.58 / 1.88 | |
| 4 | 25.84 / 1.73 | 26.31 / 1.85 | 25.05 / 2.69 | 26.36 / 3.27 | |
| 6 | 26.08 / 1.76 | 26.26 / 1.90 | 24.69 / 2.98 | 26.18 / 4.18 | |
| 8 | 25.82 / 1.76 | 26.25 / 1.91 | 24.27 / 3.01 | 26.11 / 4.69 | |
| 10 | 25.69 / 1.74 | 26.34 / 1.90 | 23.51 / 2.87 | 25.60 / 4.95 | |

Table 1: Results on the newstest2013 development set for English-German translation. Each cell lists BLEU score and mean accepted block size. Larger BLEU scores indicate higher translation quality, and larger mean accepted block sizes indicate fewer decoding iterations. The data from the table is also visually depicted in a scatter plot on the right.

From these results, we make several observations. For the regular setup with gold training data and frozen baseline model parameters, the mean block size reaches a peak of 1.76, showing that speed can be improved without sacrificing model quality. When we instead use distilled data, the BLEU score at the same block size increases by 0.43 and the mean block size reaches 1.91, showing slight improvements on both metrics. Next, comparing the results in the first two columns to their counterparts with parameter fine tuning in the last two columns, we see large increases in mean block size, albeit at the expense of some performance for larger $k$. The use of distilled data lessens the severity of the performance drop and allows for more accurate forward prediction, lending credence to our earlier intuition. The model with the highest mean block size of 4.95 is only 0.81 BLEU points worse than the initial model trained on distilled data.

We visualize the trade-off between BLEU score and mean block size in the plot next to Table 1. For both the original (●, ▲) and the distilled (■, ◆) training data, one can select a setting that optimizes for highest quality, fastest speed, or something in between. Quality degradation for larger $k$ is much less pronounced when distilled data is used. The smooth frontier in both cases gives practitioners the option to choose a setting that best suits their needs.

We also repeat the experiments from the last column of Table 1 using the top-$k$ approximate selection criterion of Section 5.1. For top-2 approximate decoding, we obtain the results $k = 2$: 26.49 / 1.92, $k = 4$: 26.22 / 3.47, $k = 6$: 25.90 / 4.59, $k = 8$: 25.71 / 5.34, $k = 10$: 25.04 / 5.67, demonstrating additional gains in accepted block size at the cost of further decrease in BLEU. Results for top-3 approximate decoding follow a similar trend: $k = 2$: 26.41 / 1.93, $k = 4$: 26.14 / 3.52, $k = 6$: 25.56 / 4.69, $k = 8$: 25.41 / 5.52, $k = 10$: 24.68 / 5.91. On the other hand, experiments using a minimum block size of $k = 2$ or $k = 3$ as described in Section 5.3 exhibit much larger drops in BLEU score with only minor improvements in mean accepted block size, suggesting that the ability to accept just one token on occasion is important and that a hard lower bound is somewhat less effective.

## 7.2 Image Super-Resolution

For our super-resolution experiments, we use the training and development data from the CelebA dataset (Liu et al., 2015). Our task is to generate a $32 \times 32$ pixel output image from an $8 \times 8$ pixel input. Our baseline model is an Image Transformer (Parmar et al., 2018) with 1D local attention trained for 1,000,000 steps on 8 P100 GPUs using the `img2img_transformer_b3` hyperparameter set. As with our machine translation experiments, we train a collection of additional models with warm-started parameters for various block sizes $k$, both with and without fine tuning of the base model's parameters. Here we train for an additional 250,000 steps.

We measure the mean accepted block size on the development set for each model. For the Image Transformer, an image is decomposed into a sequence of red, green, and blue intensities for each pixel in raster scan order, so each output token is an integer between 0 and 255. During inference, we either require an exact match with the greedy model or allow for an approximate match using the distance-based selection criterion from Section 5.2 with $\epsilon = 2$. Our results are shown in Table 2.

| $k$ | Regular | Approximate | Fine Tuning | Both |
|---|---|---|---|---|
| 1 | 1.00 | | | |
| 2 | 1.07 | 1.24 | 1.59 | 1.96 |
| 4 | 1.08 | 1.36 | 2.11 | 3.75 |
| 6 | 1.09 | 1.38 | 2.23 | 5.25 |
| 8 | 1.09 | 1.49 | 2.17 | 6.36 |
| 10 | 1.10 | 1.40 | 2.04 | 6.79 |

Table 2: Results on the CelebA development set. Each cell lists the mean accepted block size during decoding; larger values indicate fewer decoding iterations.

We find that exact-match decoding for the models trained with frozen base parameters is perhaps overly stringent, barely allowing for any speedup for even the largest block size. Relaxing the acceptance criterion helps a small amount, though the mean accepted block size remains below 1.5 in all cases. The models with fine-tuned parameters fare somewhat better when exact-match decoding is used, achieving a mean block size of slightly over 2.2 in the best case. Finally, combining approximate decoding with fine tuning yields results that are substantially better than when either modification is applied on its own. For the smaller block sizes, we see mean accepted block sizes very close to the maximum achievable bound of $k$. For the largest block size of 10, the mean accepted block size reaches an impressive 6.79, indicating a nearly 7x reduction in decoding iterations.

To evaluate the quality of our results, we also ran a human evaluation in which workers on Mechanical Turk were shown pairs of decoder outputs for examples from the development set and were asked to pick which one they thought was more likely to have been taken by a camera. Within each pair, one image was produced from the model trained with $k = 1$ and frozen base parameters, and one image was produced from a model trained with $k > 1$ and fine-tuned base parameters. The images within each pair were generated from the same underlying input, and were randomly permuted to avoid bias. Results are given in Table 3.

| Method 1 | Method 2 | 1 > 2 | Confidence Interval |
|---|---|---|---|
| Fine tuning, exact, $k = 2$ | Regular, exact, $k = 1$ | 52.8% | (50.8%, 54.9%) |
| Fine tuning, exact, $k = 4$ | Regular, exact, $k = 1$ | 54.4% | (52.5%, 56.3%) |
| Fine tuning, exact, $k = 6$ | Regular, exact, $k = 1$ | 53.2% | (51.3%, 55.0%) |
| Fine tuning, exact, $k = 8$ | Regular, exact, $k = 1$ | 55.1% | (53.3%, 56.8%) |
| Fine tuning, exact, $k = 10$ | Regular, exact, $k = 1$ | 54.5% | (53.1%, 56.0%) |
| Fine tuning, approximate, $k = 2$ | Regular, exact, $k = 1$ | 50.0% | (48.4%, 51.5%) |
| Fine tuning, approximate, $k = 4$ | Regular, exact, $k = 1$ | 53.3% | (51.7%, 55.0%) |
| Fine tuning, approximate, $k = 6$ | Regular, exact, $k = 1$ | 56.8% | (55.4%, 58.2%) |
| Fine tuning, approximate, $k = 8$ | Regular, exact, $k = 1$ | 55.2% | (53.5%, 56.7%) |
| Fine tuning, approximate, $k = 10$ | Regular, exact, $k = 1$ | 50.3% | (48.9%, 51.8%) |

Table 3: Human evaluation results on the CelebA development set. In each row, we report the percentage of votes cast in favor of the output from Method 1 over that of Method 2, along with a 90% bootstrap confidence interval.

In all cases we obtain preference percentages close to 50%, indicating little difference in perceived quality. In fact, subjects generally showed a weak preference toward images generated using the fine-tuned models, with images coming from a fine-tuned model with approximate decoding and a medium block size of $k = 6$ obtaining the highest scores overall. We believe that the more difficult training task and approximate acceptance criterion both helped lead to outputs with slightly more noise and variation, giving them a more natural appearance when compared to the smoothed outputs that result from the baseline. See Section 7.4 for examples.

## 7.3 Wall-Clock Speedup

So far we have framed our results in terms of the mean accepted block size, which is reflective of the speedup achieved relative to greedy decoding in terms of number of decoding iterations. Another metric of interest is actual wall-clock speedup relative to greedy decoding, which takes into account the additional overhead required for blockwise parallel prediction. We plot these two quantities against each other for the best translation and super-resolution settings in Figure 4.

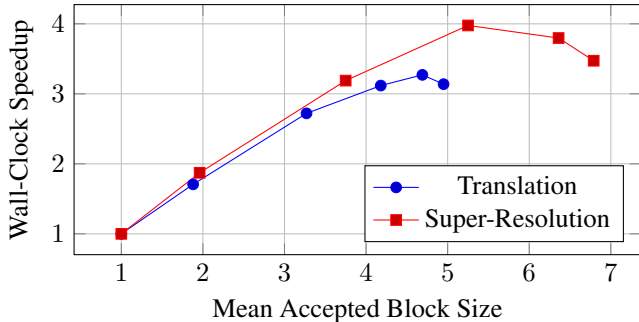

Figure 4: A plot of the relative wall-clock speedup achieved for various mean accepted block sizes, where the latter measure a reduction in iterations required for decoding. This data comes from the final column of Table 1 (translation results using fine tuning and distillation) and the final column of Table 2 (super-resolution results using fine tuning and approximate decoding).

For translation, the wall-clock speedup peaks at 3.3x, corresponding to the setting with $k = 8$ and mean accepted block size 4.7. For super-resolution, the wall-clock speedup reaches 4.0x, corresponding to the setting with $k = 6$ and mean accepted block size 5.3. In both cases, larger block sizes $k$ continue to improve in terms of iteration count, but start to decline in terms of wall-clock improvement due to their higher computational cost.

Using our best settings for machine translation (distilled data and fine-tuned models), we also ran a test set evaluation on the newstest2014 dataset. These results along with others from related approaches are summarized in Table 4. Our technique exhibits much less quality degradation relative to our baseline when compared with other approaches, demonstrating its efficacy for faster decoding with minimal impact on end performance.

## 7.4 Examples

**Machine translation.** Here we show the generation process for a typical machine translation example. Generation occurs at the level of subwords, with underscores indicating word boundaries.

*Input:* The James Webb Space Telescope (JWST) will be launched into space on board an Ariane5 rocket by 2018 at the earliest.

*Output:* Das James Webb Space Teleskop (JWST) wird bis spätestens 2018 an Bord einer Ariane5-Rakete in den Weltraum gestartet.

```
Step 1    10 tokens    [Das_, James_, Web, b_, Space_, Tele, sko, p_, (_, J]
Step 2     5 tokens    [W, ST_, ) _, wird_, bis_]
Step 3     4 tokens    [späte, stens_, 2018_, an_]
Step 4    10 tokens    [Bord_, einer_, Ari, ane, 5_, -_, Rak, ete_, in_, den_]
Step 5     2 tokens    [Weltraum, _]
Step 6     3 tokens    [gestartet_, ._, <EOS>]
```

| Model | Source | BLEU | Wall-Clock Speedup |
|---|---|---|---|
| Transformer (beam size 4) | Vaswani et al. (2017) | 28.4 | |
| Transformer (beam size 1) | Gu et al. (2018) | 22.71 | |
| Transformer (beam size 4) | Gu et al. (2018) | 23.45 | |
| Non-autoregressive Transformer | Gu et al. (2018) | 17.35 | |
| Non-autoregressive Transformer (+FT) | Gu et al. (2018) | 17.69 | |
| Non-autoregressive Transformer (+FT + NPD $s = 10$) | Gu et al. (2018) | 18.66 | |
| Non-autoregressive Transformer (+FT + NPD $s = 100$) | Gu et al. (2018) | 19.17 | |
| Transformer (beam size 1) | Lee et al. (2018) | 23.77 | 1.20x |
| Transformer (beam size 4) | Lee et al. (2018) | 24.57 | 1.00x |
| Iterative refinement Transformer ($i_{dec} = 1$) | Lee et al. (2018) | 13.91 | 11.39x |
| Iterative refinement Transformer ($i_{dec} = 2$) | Lee et al. (2018) | 16.95 | 8.77x |
| Iterative refinement Transformer ($i_{dec} = 5$) | Lee et al. (2018) | 20.26 | 3.11x |
| Iterative refinement Transformer ($i_{dec} = 10$) | Lee et al. (2018) | 21.61 | 2.01x |
| Iterative refinement Transformer (Adaptive) | Lee et al. (2018) | 21.54 | 2.39x |
| Latent Transformer without rescoring | Kaiser et al. (2018) | 19.8 | |
| Latent Transformer rescoring top-10 | Kaiser et al. (2018) | 21.0 | |
| Latent Transformer rescoring top-100 | Kaiser et al. (2018) | 22.5 | |
| Transformer with distillation (greedy, $k = 1$) | This work | 29.11 | 1.00x |
| Blockwise parallel decoding for Transformer ($k = 2$) | This work | 28.95 | 1.72x |
| Blockwise parallel decoding for Transformer ($k = 4$) | This work | 28.54 | 2.69x |
| Blockwise parallel decoding for Transformer ($k = 6$) | This work | 28.11 | 3.10x |
| Blockwise parallel decoding for Transformer ($k = 8$) | This work | 27.88 | 3.31x |
| Blockwise parallel decoding for Transformer ($k = 10$) | This work | 27.40 | 3.04x |

Table 4: A comparison of results on the newstest2014 test set for English-German translation. The reported speedups are for wall-clock time for single-sentence decoding averaged over the test set. Our approach exhibits relatively little loss in quality compared to prior work. We achieve a BLEU score within 0.29 of the original Transformer with a real-time speedup over our baseline exceeding 3x.

**Super-resolution.** Here we provide a selection of typical examples from the development set. As suggested by the human evaluations in Section 7.2, the blockwise parallel decodes are largely comparable in quality to the standard greedy decodes. For each triple, the left image is the low-resolution input, the middle image is the standard greedy decode, and the right image is the approximate greedy decode using the fine-tuned model with block size $k = 10$.

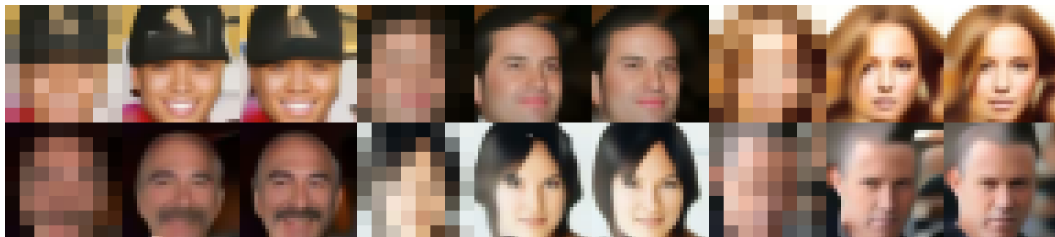

# 8 Conclusion

In this work, we proposed blockwise parallel decoding as a simple and generic technique for improving decoding performance in deep autoregressive models whose architectures allow for parallelization of scoring across output positions. It is comparatively straightforward to add to existing models, and we demonstrate significant improvements in decoding speed on machine translation and a conditional image generation task at no loss or only small losses in quality.

In future work we plan to investigate combinations of this technique with potentially orthogonal approaches such as those based on sequences of discrete latent variables (Kaiser et al., 2018).

## Acknowledgments

We thank Arvind Neelakantan, Niki Parmar, and Ashish Vaswani for their generous assistance in setting up the Image Transformer and distillation experiments.

## Footnotes

[1]The BLEU scores in the first two columns vary slightly with $k$. This is because the final decoder layer is processed by a learned transformation for all predictions $p_1, p_2, \ldots, p_k$ in our implementation rather than just $p_2, \ldots, p_k$. Using an identity transformation for $p_1$ instead would result in identical BLEU scores.

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
