[Reviews · NeurIPS 2018]

Reviewer 1



This paper presents a parallel inference algorithm that allows performing greedy decoding in sub-linear time. The proposed approach considers using k auxiliary models, in which each one predicts the j-th next word. In this way, these k models can run and predict the next k words in parallel. Then a verification step is conducted to verify the prediction of these k models. If the proposal is correct, then the predictions are accepted and the decoding can move on. In this way, in the best case, the model only requires m/k step to generate a sequence with m tokens. Pros: - The idea proposed in this paper is neat and interesting. - The authors present experiments on tasks in two different domains to empirically demonstrate the efficiency of the decoding algorithm. - The authors proposed both exact and inexact algorithms for making the predictions. - The authors provide a nice analysis of their approach. Cons: - The writing of the paper can be improved, and the description of the proposed approach is sometimes confusing and descriptions of experimental settings and results are handwaving. - The speed-up of the decoding is not guaranteed and require large duplicated computation, although the computation can be amortized. Comments: - It is unclear what the knowledge distillation approach is actually used in the experiment. - I do not fully understand Table 1. It is unclear if the results showing is the exact or the approximate inference approach. It seems that the results showing here is using the exact greedy approach as the authors called it regular. But, in this case, why the Blue score is different for different k? If I understand correctly the p_1 model for different k should be the same, so the exact decoding should give the same output. - Line 210-214: for approaches except "Regular", the authors should quantify the quality of the decoding quantitively. For example, what is the mean difference in pixel values between images generated by approximation greedy decoding and exact greedy decoding? How many of them can the difference be perceived by a human? Minor comments: - Figure 2 is not readable when the paper is printed in white and black. - Line 109: Figure 3 is not showing what the authors describe. - Line 199-120: Please explicitly define what do you mean by "Regular" and "Approximate" in Table 2. ----- Comments after rebuttal: Thanks for the clarification and additional experiments on showing performance measured under wall-clock time and Table 1. With the clarification and the results on wall-clock, I update my rating.

Reviewer 2



The paper presents a simple technique for speeding up the decoding process for a generic autoregressive seq2seq model. The proposed technique consists of some heuristics-based methods and it looks very straightforward to me and involves little novelty. Pros: The authors have presented a technique for speeding up the decoding process for deep autoregressive models. The empirical results demonstrated the effectiveness. Cons: The technique involves some heuristics-based methods. The top-k and distance-based methods are used, but these are very typical techniques used in the literature. It seems there is very little novelty involved.

Reviewer 3



The authors propose to speed up decoding by taking advantage of parallel generations via Transformer. Specifically, by having models predict the kth next word in parallel and then validating those predictions in parallel, they are able to achieve a speed of 2m/k for a sequence of length m and prediction of up to k. This is further reduced to m/k+1 via combining proposal and verification steps. The primary use case is presumably for single sentence decoding because batch decoding won't have spare compute. The result I find most surprising is that the models perform better with k>1. Can this be justified better? Can Fig 3 be augmented with or duplicated for MT? My main concern is that I don't have a great understanding of where this technique fails? Is there something interesting about the predictions that are shorter? For example, for MT how much of the speed gains are just from predictable stop words or selectional preferences? Given that there is basically an entire free page, I'd really like to have 1. Examples be provided for Super-Resolution since there is no direct evaluation 2. Examples of how Both succeeds for ave k > 4 while regular/distillation fail.